# Establishment of a Prediction Model for Overall Survival after Stereotactic Body Radiation Therapy for Primary Non-Small Cell Lung Cancer Using Radiomics Analysis

**DOI:** 10.3390/cancers14163859

**Published:** 2022-08-10

**Authors:** Subaru Sawayanagi, Hideomi Yamashita, Yuki Nozawa, Ryosuke Takenaka, Yosuke Miki, Kosuke Morishima, Hiroyuki Ueno, Takeshi Ohta, Atsuto Katano

**Affiliations:** Department of Radiology, University of Tokyo Hospital, 7-3-1, Hongo, Bunkyo-ku, Tokyo 113-8655, Japan

**Keywords:** non-small cell lung cancer, stereotactic body radiation therapy, radiomics, overall survival

## Abstract

**Simple Summary:**

Lung cancer remains the leading cause of cancer-related mortality worldwide. Although early-stage non-small cell lung cancer (NSCLC) is likely to be controlled with stereotactic body radiation therapy (SBRT), approximately 18% of patients lead to recurrence. The aim of this study was to identify prognostic factors and establish a predictive model for survival outcomes of patients with non-metastatic NSCLC treated with SBRT. Several radiomic features were selected as predictive factors and two prediction models were established from the pre-treatment computed tomography images of 250 patients in the training cohort. One radiomic factor remained a significant prognostic factor of overall survival (OS) (*p* = 0.044), and one predicting model could estimate OS time (mean: 37.8 months) similar to the real OS time (33.7 months). In this study, we identified one radiomic factor and one prediction model that can be widely used.

**Abstract:**

Stereotactic body radiation therapy (SBRT) for early-stage non-small cell lung cancer (NSCLC) leads to recurrence in approximately 18% of patients. We aimed to extract the radiomic features, with which we predicted clinical outcomes and to establish predictive models. Patients with primary non-metastatic NSCLC who were treated with SBRT between 2002 and 2022 were retrospectively reviewed. The 358 primary tumors were randomly divided into a training cohort of 250 tumors and a validation cohort of 108 tumors. Clinical features and 744 radiomic features derived from primary tumor delineation on pre-treatment computed tomography were examined as prognostic factors of survival outcomes by univariate and multivariate analyses in the training cohort. Predictive models of survival outcomes were established from the results of the multivariate analysis in the training cohort. The selected radiomic features and prediction models were tested in a validation cohort. We found that one radiomic feature showed a significant difference in overall survival (OS) in the validation cohort (*p* = 0.044) and one predicting model could estimate OS time (mean: 37.8 months) similar to the real OS time (33.7 months). In this study, we identified one radiomic factor and one prediction model that can be widely used.

## 1. Introduction

Lung cancer is one of the most common causes of cancer-related mortality worldwide [1,2,3]. Non-small cell lung cancer (NSCLC) is the major subtype of lung cancer, accounting for approximately 85% of all lung cancer cases [4]. Stereotactic body radiation therapy (SBRT) is currently one of the standard curative treatment options for patients with early-stage NSCLC who are medically inoperable or refuse surgery [5,6,7,8,9]. While a very high local tumor control rate of 92–98% can be expected [10,11,12,13,14], SBRT for early-stage NSCLC leads to local, regional, or distant metastatic recurrence in 18–20% of cases [13,14,15,16].

Appropriate prediction of recurrence risk before treatment is essential for personalized treatment. If we can select patients with a high possibility of recurrence in advance, we may increase the treatment intensity by, for example, changing the total radiation dose and fractions. It has been reported that mediastinal pleural contact, maximum tumor diameter, and the maximum standard uptake value (SUV-max) in positron emission tomography (PET) are associated with distant metastasis as clinical features in patients with stage I-II NSCLC treated with SBRT [17]. Other groups have reported that the biologically effective dose (BED) is also related to the prognosis of patients treated with SBRT for early-stage NSCLC [18]. Clinical T stage, tumor size, and total radiation dose also had a statistically significant impact on survival outcomes of patients with early-stage NSCLC treated with SBRT [19,20]. Nowadays, many quantitative features can be extracted from radiographic images using radiomics. Radiomic features potentially include information about tumor heterogeneity, tumor microenvironment, and underlying gene-expression patterns, which may be associated with prognosis [21,22]. Radiomics approaches are widely used to detect prognostic factors in several types of cancers, such as hepatic cancer, brain tumor, pancreatic cancer, and head and neck cancer [21,23,24,25,26]. Several articles have reported prognostic factors in patients with early-stage NSCLC who underwent SBRT using radiomics [27,28,29,30,31,32,33]. They concluded that some radiomic features were prognostic factors and showed predictive models of prognosis; however, there is currently no widely used prediction model. It is possible that the number of patients, which was approximately 100, was not large enough to establish a versatile prediction model.

## 2. Materials and Methods

### 2.1. Patients

This study was reviewed and approved by the institutional review board and ethics committee (examination number 3372). We retrospectively reviewed an institutional database to identify patients with primary non-metastatic NSCLC who were treated with SBRT with curative intent between April 2002 and March 2022 with or without salvage treatment. Patients who had plain chest CT scans within two weeks prior to SBRT were included. Patients who had never been examined to confirm the clinical outcomes after SBRT were excluded. We detected 358 primary tumors in 338 patients. A total of 250 primary tumors were randomly assigned to the training cohort, and 108 tumors were assigned to the validation cohort.

### 2.2. Radiotherapy

All patients with no pathological diagnosis by biopsy required a continuous increase in the primary tumor size or abnormal accumulation of fluorodeoxyglucose in the tumor before SBRT. Tumor markers (neuron-specific enolase, NSE, and pro-gastrin-releasing peptide, ProGRP) needed to be checked to exclude small cell carcinoma. The patient’s forced expiratory volume in one second needed to be more than 750 cc to receive SBRT. In addition, indications for SBRT must be decided by an institutional cancer board consisting of pulmonologists, thoracic surgeons, radiologists, and radiation oncologists.

Six or 10 MV photon linear accelerators were used for SBRT. The radiation technique used was three-dimensional conformal radiotherapy (3D-CRT) or volumetric modulated arc therapy (VMAT). The fixed multi-port technique consisting of 7–12 beams was used as 3D-CRT. A total dose of 42–64 Gy in 4–10 fractions was administered. When the α/β ratio is assumed to be 10, BED was 75–166 Gy. The total dose and fractions were determined by radiation oncologists by considering the location of the tumor and the expected dose of organs at risk. Gross tumor volume (GTV) was contoured on the planning CT images displayed at a lung window level. The clinical target volume (CTV) was defined to be the same as the GTV. An internal margin was added to the CTV to cover the respiratory motion of the tumor and generate the internal target volume (ITV) using four-dimensional CT images. A tracking system was not used. Then, a 5 mm margin in every direction was added to the ITV to generate the planning target volume (PTV). Radiation dose was prescribed to 95% of the PTV, and radiation was administered every weekday.

### 2.3. Clinical Endpoints

As background features, age, Karnofsky performance status (KPS), sex, total radiation dose, radiotherapy (RT) technique (fixed multi-ports, VMAT, or both), primary tumor location, maximum tumor diameter, and SUV-max in PET were investigated.

Overall survival (OS), local relapse-free survival (LRFS), and progression-free survival (PFS) were used as the endpoints. These were counted from the date of SBRT initiation. The patients were usually followed-up and underwent a CT scan every 2–3 months for the first year and repeated 3–6 months thereafter. Local recurrence, which meant a recurrence in the radiation field, was usually identified on PET when SUV-max of the tumor was higher than 2.5 or by biopsy in case of obvious enlargement of the local tumor on the follow-up CT scan. However, some relapses were judged only by continuous enlargement on follow-up CT.

### 2.4. CT Image and Tumor Contouring

All patients underwent plain chest CT for treatment planning. We used Aquilion LB (Canon Medical Systems Corporation, Otawara, Japan) as the CT scanner. The CT tube voltage was 120 kV, the tube current was 350 mA, and the slice thickness was 2 mm. Every primary tumor was manually delineated as the GTV for this study on the expiratory phase of the planning CT displayed with a lung window level using Monaco (Elekta AB, Stockholm, Sweden) by a single expert radiation oncologist. Tumor outline was contoured, and pleura, vessels, heart, bones, and chest wall were excluded from the GTV.

### 2.5. Radiomic Feature Extraction

We used PyRadiomics v3.0.1 (Boston, MA, USA) to extract radiomic features from contoured regions of interest (ROIs) of GTVs [34]. A fixed bin count of 64 was used for discretization of the image gray level. All CT voxels were resampled to 1 × 1 × 1 mm^3^ using a B-Spline interpolation function. A total of 107 quantitative features were automatically extracted, including 19 first-order statistics features (intensity histogram, IH), 26 shape-based histogram features, and texture features (gray-level co-occurrence matrix, GLCM, 24 features; gray-level run-length matrix, GLRLM, 16 features; gray-level size-zone matrix, GLSZM, 16 features; neighboring gray-tone difference matrix, NGTDM, 5 features; and gray-level dependence matrix, GLDM, 14 features). All quantitative features are listed in Appendix A. From these original radiomic features, 744 quantitative features were obtained using wavelet transformation. Coiflets 1, which is one of the methods of wavelet transformation, was used in x, y and z axes in this study. As a result, 8 wavelet-transformed images (HHH, HHL, HLH, HLL, LHH, LHL, LLH, and LLL) were obtained, where H represents high frequency and L represents low frequency. Mathematical definitions of these radiomic features have previously been described and are available at https://pyradiomics.readthedocs.io/en/latest/features.html (accessed on 11 June 2022).

### 2.6. Statistical Analysis

The Kaplan–Meier method was used for survival time analysis. Clinical features of continuous variables consisted of maximum tumor diameter, age, KPS, SUV-max in PET, and total radiation dose, and radiomic features were divided into two groups according to the median values in the training cohort. Univariate analysis was performed using the log-rank test to assess the difference in clinical outcomes between the high- and low-value groups. Multivariate analysis was conducted using the Cox proportional hazards model among the factors with a significant difference in univariate analysis. The stepwise method with the Bayesian information criterion was used for variable selection in the training cohort (direction = “backward/forward”). We created a prediction score from the results of the Cox proportional hazards model and multiple linear regression analysis of the training cohort. The selected radiomic factors and prediction scores were validated in the validation cohort. All statistical analyses were performed using R v4.1.3. The results of the univariate analysis of radiomic features were considered statistically significant at *p* < 0.01, whereas the other results were considered statistically significant at *p* < 0.05.

## 3. Results

### 3.1. Patient Characteristics

Patients were randomly divided into training (n = 250) and validation (n = 108) groups. As shown in Table 1, there was no bias in the proportions or means of both the patient and tumor background factors in either group. In the training group, 68% were male, mean age was 77 (range, 42–93) years, 84% had peripheral lesions, mean maximum tumor diameter was 20.5 mm (4.0–90.0), mean total dose was 52.5 Gy (42–64), 70% received VMAT, and 12.8% received salvage treatment after recurrence was recognized. No patient underwent adjuvant treatment after SBRT. The median follow-up time for patients who survived at the last follow-up of the training cohort and the validation cohort was 33.0 (0.9–167.9) and 33.9 (0.5–149.4) months, respectively.

The median crude survival of the training cohort and the validation cohort was 30.4 (interquartile range (IQR), 15.9–50.9) and 28.2 (14.2–44.1) months, respectively. Similarly, the median crude LRFS was 24.6 (10.4–45.8) vs. 23.2 (11.4–38.1) months and the median crude PFS was 23.3 (9.7–45.0) vs. 21.3 (11.0–38.4).

### 3.2. Univariate Analysis

As shown in Table 2, as background factors affecting OS, KPS, maximum tumor diameter, sex, peripheral lesions, and salvage treatment showed a significant difference of *p* < 0.05 in the univariate analysis using the log-rank test. Regarding LRFS, KPS, maximum tumor diameter, sex, peripheral lesions, consolidation/tumor (C/T) ratio, salvage treatment, and SUV-max showed a significant difference of *p* < 0.05 in the univariate analysis. For PFS, KPS, peripheral lesions, and salvage treatment showed a significant difference of *p* < 0.05 in the univariate analysis.

Among the 744 radiomic features, 70 had a significant difference of *p* < 0.01 in OS between the high- and low-value groups in the training cohort in the log-rank test. For LRFS, 223 factors, and for PFS, 210 factors showed a significant difference of *p* < 0.01.

### 3.3. Multivariate Analysis

As shown in Table 3, multiple factors with significant differences in univariate analysis were subjected to the Cox proportional hazards regression model. As a result, four factors, (1) “90 Percentile_HHH,” (2) “LargeAreaEmphasis_LHH,” (3) “Mean_HHH,” and (4) “Median_HLL” remained as factors related to OS. Similarly, regarding LRFS, two factors, (1) “InverseVariance_HLL” and (2) “SmallDependenceHighGrayLevelEmphasis_HHH” remained. Regarding PFS, three factors, (1) “SmallDependenceHighGrayLevelEmphasis_HHH,” (2) “TotalEnergy_HHL,” and (3) “JointEntropy_HLL” remained.

### 3.4. Prediction Score

From the results of the Cox proportional hazards regression model, we created a prediction score, which was named the Cox-score, calculated by the following formula using the coef value as a coefficient.

Cox-scoreOS = −0.6545 × [90 Percentile_HHH] + 0.5871 × [LargeAreaEmphasis_LHH] + 0.6795 × [Mean_HHH] + 0.6852 × [Median_HLL]

Cox-scoreLRFS = 1.2655 × [InverseVariance_HLL] − 1.5453 × [SmallDependenceHighGrayLevelEmphasis_HHH]

Cox-scorePFS = −0.9901 × [InverseVariance_HLL] + 0.9006 × [InverseVariance_HLL] − 0.6994 × [InverseVariance_HLL]

For the [x] value, “+1” is substituted if the value of x is higher than the median value, and “−1” is substituted if the value is lower than the median value.

We estimated survival time using multiple linear regression analysis in the training cohort. The estimated survival time was calculated using the following formula.

Estimated OS time (months) = 49.128 + 9.528 × [90 Percentile_HHH] − 11.909 × [LargeAreaEmphasis_LHH] − 6.781 × [Mean_HHH] − 11.658 × [Median_HLL]

Estimated LRFS time (months) = 61.431 − 22.649 × [InverseVariance_HLL] + 31.459 × [SmallDependenceHighGrayLevelEmphasis_HHH]

Estimated PFS time (months) = 38.400 + 8.804 × [JointEntropy_HLL] + 21.647 × [SmallDependenceHighGrayLevelEmphasis_HHH] − 14.024 × [TotalEnergy_HHL]

For the [x] value, “+1” is substituted if the value of x is higher than the median value, and “0” is substituted if the value is lower than the median value. 

### 3.5. Validation

As a result of the log-rank test of 108 cases of the validation cohort for each of the radiomic factors remaining in the multivariate analysis, only one factor of “LargeAreaEmphasis_LHH” showed a significant difference in OS (*p* = 0.044, 5-year OS of 70.7% vs. 50.3%) (Figure 1). “LargeAreaEmphasis_LHH” was a parameter that was categorized in GLSZM features. Regarding LRFS and PFS, none of the factors that showed a significant difference in the multivariate analysis of the training cohort showed a significant difference in the univariate analysis of the validation cohort.

Regarding the Cox-score, the value for each case was calculated using the distribution of the radiomic features of 108 cases in the validation group. As a result, 38 cases with a score of less than −0.1 were divided into a low value group, 36 cases with a score of +1.0 or more were divided into a high value group, and the other 34 cases were divided into a median value group for OS. In the log-rank test, the difference in OS between the three groups was not significant (*p* = 0.63, 5-year OS of 62.6% in the high score group, 59.6% in the median, and 65.4% in the low score group). The difference in LRFS between the two groups, if +1.0 was the cutoff value, was not significant (*p* = 0.086, 5-year OS of 60.8% in the high, and 82.8% in the low). The difference in PFS between the two groups, if −1.0 was the cutoff value, was not significant (*p* = 0.070, 5-year OS of 52.0% in the high, and 94.7% in the low). 

We compared the estimated survival time calculated using the formula from multiple linear regression analysis with the real survival time in the validation cohort. As a result of the estimated OS time, the difference in the average values was 4.1 months (33.7 months in the real survival time vs. 37.8 months in the estimated survival time) and the t value and *p* value were 1.619 and 0.11 by the paired *t*-test, respectively (Figure 2). Regarding LRFS and PFS, the *p* value was less than 0.0001 using the paired *t*-test.

## 4. Discussion

In this study, we extracted radiomic features from the GTV on SBRT planning CT of patients with non-metastatic primary NSCLC and selected features correlated with survival outcomes. We established a prediction model using these selected features. The selected features and prediction models were validated in an independent external cohort. “LargeAreaEmphasis_LHH” categorized in GLSZM features remained as a factor significantly correlated with OS in the validation cohort. The estimated OS time calculated using the formula from the multiple linear regression analysis was similar to the actual OS time in the validation cohort. There was no correlation between the radiomic features and LRFS or PFS.

Prediction of recurrence risk before treatment for early-stage NSCLC is so important. The treatment intensity such as the total radiation dose, fractions, and radiation field can be increased for high-risk patients. Another measure to increase the treatment intensity is combined therapy. Ernani et al. [35] showed the promising OS data of SBRT and adjuvant chemotherapy for patients with tumors less than 4 cm and node-negative NSCLC. Use of immune checkpoint inhibitors as an adjuvant therapy after curative chemoradiotherapy for advanced stage NSCLC have rapidly spread worldwide [36,37]. Several studies have also reported the effectiveness of immune checkpoint inhibitors together with SBRT for high-risk patients with early-stage NSCLC [38]. The selected radiomic features and OS prediction model derive in the present study may contribute to selecting patients who benefit from these treatment options.

Several studies have investigated the prognostic factors of patients who received SBRT for early-stage NSCLC using radiomics analysis. Oikonomou et al. [32] examined CT and PET-derived radiomic features in 150 patients with non-metastatic NSCLC treated with SBRT. They examined several principal components, consisting of six to eight radiomic features. Four principal components were found to be significantly associated with the survival outcomes. In the combined analysis with SUV-max, radiomic features remained the only predictors of OS, disease specific survival, and regional control. Huynh et al. [30] analyzed 113 patients with stage I–II NSCLC who were treated with SBRT. Among the 1605 radiomic features extracted from the tumors in the free breathing CT images, one radiomic feature was a significant prognostic factor for distant metastasis and four radiomic features were prognostic for OS. Starkov et al. [33] retrospectively analyzed 116 patients treated with SBRT for biopsy-confirmed primary NSCLC at a single institution. They showed that quantitative image features from NSCLC nodules on pre-treatment CT images were correlated with OS after SBRT. Similar to the present study, no significant relationship was observed between radiomic features and survival outcomes except for OS. No external validation cohort was established in any of these studies. In contrast, Dissaux et al. [28] built a prediction model of local control derived from PET among a training set of 64 early-stage NSCLC patients treated with SBRT. The model maintained a significant correlation with local control in the validation set of 23 patients. The fact that patients in the training set were collected from two institutions and those in the validation set were collected from another single institution may have affected the results of the study. Yu et al. [39] retrospectively extracted radiomic features from contrast-enhanced CT images of a training cohort (n = 147) and an independent validation cohort (n = 295) of patients with stage I NSCLC. Although the cohort size was large, there was an important difference between the two cohorts. The prediction model consisting of two radiomic features was significantly associated with both OS and distant metastasis in the validation cohort. Patients in the training cohort were treated with surgery, and those in the validation cohort were treated with SBRT. In addition, the patients in the validation cohort were significantly older than those in the training cohort, which seemed to be due to the difference in the indication for treatment. These differences may have affected the accuracy of the validation. In contrast, the training and validation cohorts were randomly divided in the present study. Thus, the clinical background features were not significantly different between the two cohorts. Franceschini et al. [29] also analyzed radiomic features in the treatment planning CT images of 70 patients treated with SBRT for early-stage NSCLC for training and 32 patients for validation. The two cohorts were randomly divided as in the present study. A set of 45 textural features was extracted from the tumor volumes on the treatment planning CT images. They built a predictive model that was prognostic of PFS and disease-specific survival. Although these studies included a validation process, the sample size may not have been large enough. In our study, we examined much more NSCLC nodules, which were randomly divided into training and validation cohorts to minimize the risk of bias. In addition to background factors, crude OS, LRFS, and PFS were similar between the two cohorts. Several radiomic features remained as factors correlated with PFS and LRFS in the multivariate analysis of the training cohort; however, none of them were found to be prognostic factors in the external validation process. Unless validation is conducted, these radiomic features could be identified as prognostic factors.

The present study has some limitations. First, although PET or biopsy was recommended to identify local recurrence, there were some cases in which local recurrence was judged using only CT. Thus, there may have been an overdiagnosis of local recurrence. Second, all patients were recruited from a single institution and all tumors were contoured by a single radiation oncologist. External validation at various institutions is expected to be conducted to develop the established prediction model into a widely usable model. Third, the fact that all GTVs were manually contoured may have reduced reproducibility of this study. Homogeneity of this study was kept since all contouring tasks were conducted by a single expert radiation oncologist.

## 5. Conclusions

We found that the radiomic features categorized as GLSZM features derived from the GTV on the pre-treatment CT image were an independent prognostic factor of OS in non-metastatic NSCLC patients treated with curative SBRT. Our findings suggest that the estimated OS time derived from the multiple linear regression analysis was similar to the real OS time. These findings may contribute to selecting patients who benefit from increasing the treatment intensity by, for example, increasing radiation dose and using combined therapy.

The selected radiomic factor and OS prediction model derived in this study can be widely used in the future. External validation at various institutions is expected.

## Figures and Tables

**Figure 1 cancers-14-03859-f001:**
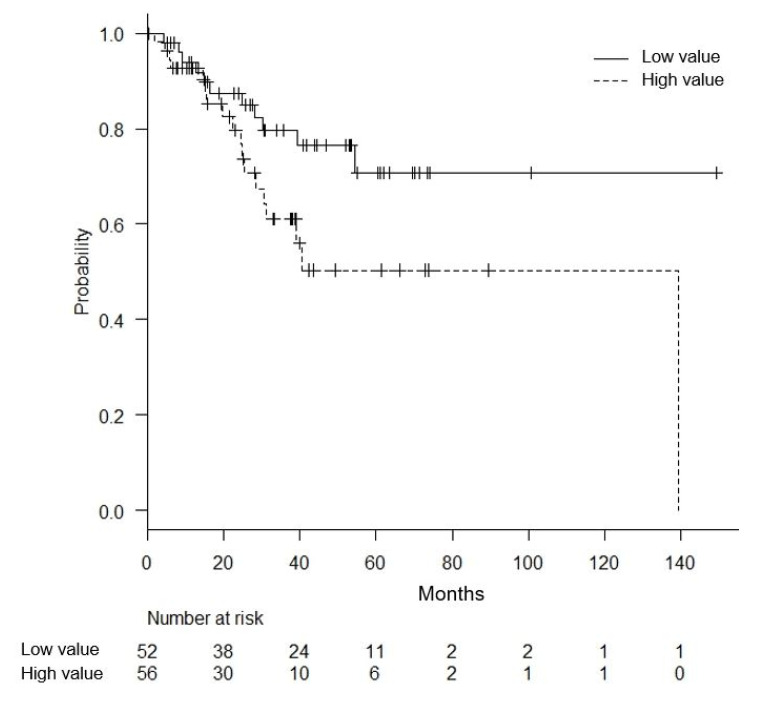
Impact of the radiomic factor named “LargeAreaEmphasis_LHH” on overall survival in the validation cohort. Patients were divided into two groups according to the median value of “LargeAreaEmphasis_LHH” (low value group and high value group). The univariate analysis using the log-rank test was conducted (*p* = 0.044, 5-year OS of 70.7% vs. 50.3%).

**Figure 2 cancers-14-03859-f002:**
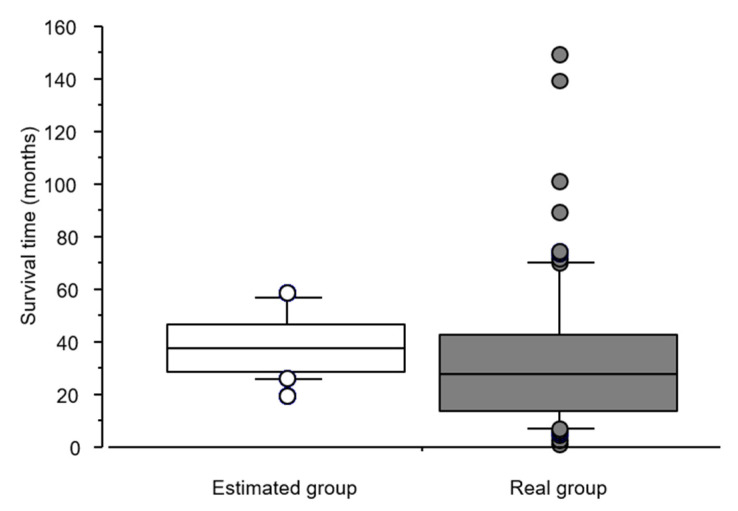
Boxplots of the estimated OS time and the real OS time in the validation cohort. The estimated survival time was calculated by the formula from the multiple linear regression analysis. with the real survival time in the validation cohort. The average estimated OS time was 33.7 months, and the average real OS time was 37.8 months (*p* = 0.11 by the paired *t*-test).

**Table 1 cancers-14-03859-t001:** Patient characteristics of the training and the validation cohort.

Factor	Training		Validation		*p* Value
	N	Rate	N	Rate	
N	250		108		
Histology					
Adenocarcinoma	47	18.8%	18	16.7%	0.98
Squamous cell carcinoma	10	4.0%	4	3.7%	
Others	15	6.0%	7	6.5%	
None	178	71.2%	79	73.1%	
Location					
Central	39	15.6%	12	11.1%	0.26
Peripheral	211	84.4%	96	88.9%	
Site					
Right upper lobe	72	28.8%	40	37.0%	0.31
Right middle lobe	20	8.0%	7	6.5%	
Right lower lobe	57	22.8%	21	19.4%	
Left upper lobe	65	26.0%	20	18.5%	
Left lower lobe	36	14.4%	20	18.5%	
C/T ratio					
≥50%	200	80.0%	89	82.4%	0.66
<50%	50	20.0%	19	17.6%	
Maximum tumor diameter					
Mean (mm)	20.5		20.8		0.85
SD	10.5		9.3		
Age					
Mean (y.o.)	77.1		77.8		0.47
SD	8.9		8.1		
KPS					
Mean (%)	89.0		88.5		0.52
SD	5.8		6.4		
SUV-max					
Mean	5.9		5.5		0.59
SD	5.1		4.2		
Total radiation dose					
Mean (Gy)	52.5		51.5		0.11
SD	4.6		7.6		
Sex					
Male	169	67.6%	76	70.4%	0.60
Female	81	32.4%	32	29.6%	
RT technique					
VMAT	171	68.4%	70	64.8%	0.23
Fixed multi-port	75	30.0%	33	30.6%	
Both	4	1.6%	5	4.6%	
Salvage treatment					
Systemic therapy	15	6.0%	9	8.3%	0.45
Radiotherapy	9	3.6%	3	2.8%	
Chemoradiotherapy	3	1.2%	0	0.0%	
Surgery	3	1.2%	0	0.0%	
Other treatment	2	0.8%	0	0.0%	
None	29	11.6%	11	10.2%	
Unknown	1	0.4%	3	2.8%	
No recurrence	188	75.20%	82	75.9%	

C/T―consolidation/tumor; KPS―Karnofsky performance status; SUV-max―maximum standard uptake value; RT―radiotherapy; VMAT―volumetric modulated arc therapy.

**Table 2 cancers-14-03859-t002:** Univariate analysis of background factors using the log-rank test.

Factor	OS			LRFS			PFS		
	5-Year OS (%)	95%CI	*p* Value	5-Year LRFS (%)	95%CI	*p* Value	5-Year PFS (%)	95%CI	*p* Value
KPS									
Low value	35.5	16.2–55.5	0.0028	63.6	40.4–79.8	0.0012	50.1	30.1–67.1	0.00053
High value	61.5	51.5–70.0		79.9	71.6–85.9		65.9	56.4–73.8	
SUV-max									
Low value	61.6	43.8–75.3	0.11	79.7	64.7–88.9	0.0090	53.1	35.9–67.6	0.11
High value	41.7	23.9–58.6		60.1	42.8–73.8		47.0	31.5–61.0	
RT technique									
VMAT	60.5	48.3–70.6	0.52	77.3	66.9–84.8	0.72	58.8	47.5–68.5	0.26
Fixed multi-port	57.0	42.5–69.1		76.2	62.7–85.3		70.2	55.8–80.7	
Total radiation dose									
Low value	53.9	40.8–65.3	0.80	70.3	57.6–79.9	0.074	58.1	45.0–69.1	0.28
High value	64.4	52.4–74.0		82.5	72.2–89.3		68.3	57.4–77.0	
Age									
Low value	63.4	51.8–73.0	0.33	79.1	68.7–86.3	0.70	63.8	51.9–73.4	0.60
High value	48.7	33.5–62.4		75.1	62.0–84.2		63.6	51.0–73.8	
Histology									
Adenocarcinoma	64.6	39.4–81.4	0.25	69.9	48.1–83.9	0.87	68.7	47.5–82.7	0.73
Squamous cell carcinoma	90.0	47.3–98.5		85.7	33.4–97.9		77.1	34.5–93.9	
Others	43.8	15.7–69.1		82.1	44.4–95.3		65.0	31.0–85.4	
None	54.9	43.8–64.7		78.7	69.7–85.3		60.8	50.1–69.9	
Maximum tumor diameter									
Low value	66.6	54.2–76.4	0.042	88.3	79.5–93.5	0.0017	68.6	56.3–78.1	0.12
High value	48.0	34.6–60.1		64.8	51.7–75.1		56.9	44.3–67.7	
Sex									
Male	52.1	40.8–62.1	0.030	70.4	59.9–78.6	0.011	59.9	48.9–69.2	0.26
Female	68.2	51.5–80.1		90.3	79.0–95.6		69.4	54.7–80.2	
Location									
Central	22.1	5.9–44.6	0.00040	58.1	34.0–76.1	0.00063	26.2	5.8–53.2	0.0011
Peripheral	63.9	54.4–72.0		80.6	72.4–86.6		69.0	60.5–76.1	
Site									
Right upper lobe	66.6	46.0–80.9	0.88	74.2	57.5–85.1	0.48	55.7	40.0–68.7	0.49
Right middle lobe	48.2	17.5–73.7		76.6	48.8–90.5		64.5	30.4–85.1	
Right lower lobe	51.5	33.1–67.2		74.6	55.4–86.5		68.2	49.7–81.1	
Left upper lobe	62.4	45.3–75.5		85.3	67.9–93.7		70.2	52.2–82.6	
Left lower lobe	53.5	30.4–72.0		75.4	54.1–87.8		60.7	35.8–78.5	
C/T ratio									
≥50%	57.8	47.6–66.6	0.62	65.6	46.2–79.4	0.042	52.6	34.1–68.1	0.11
<50%	58.2	35.9–75.0		80.6	72.1–86.7		66.3	56.6–74.4	
Salvage treatment									
Systemic therapy	40.4	10.6–69.4	0.0054	11.3	0.7–38.8	<0.0001	N/A	N/A	<0.0001
Radiotherapy	23.7	1.0–63.8		N/A	N/A		N/A	N/A	
Chemoradiotherapy	N/A	N/A		N/A	N/A		N/A	N/A	
Surgery	66.7	5.4–94.5		66.7	5.4–94.5		N/A	N/A	
Other treatment	N/A	N/A		N/A	N/A		N/A	N/A	
None	N/A	N/A		N/A	N/A		N/A	N/A	
Unknown	N/A	N/A		N/A	N/A		N/A	N/A	
No recurrence	65.9	55.6–74.3		99,2	94.7–99.9		100.0	100.0–100.0	

OS―overall survival; LRFS―local relapse-free survival; PFS―progression-free survival; KPS―Karnofsky performance status; SUV-max―maximum standard uptake value; VMAT―volumetric modulated arc therapy; N/A―not applicable.

**Table 3 cancers-14-03859-t003:** Multivariate analysis of radiomic factors using the Cox proportional hazards regression model with the stepwise method.

Factor	Hazard Ratio	Lower 95%CI	Upper 95%CI	*p* Value	Coefficient
OS					
90 Percentile_HHH	0.5197	0.3223	0.8381	0.007267	−0.6545
LargeAreaEmphasis_LHH	1.7990	1.1260	2.8750	0.014120	0.5871
Mean_HHH	1.9730	1.2240	3.1790	0.005255	0.6795
Median_HLL	1.9840	1.2400	3.1740	0.004262	0.6852
LRFS					
InverseVariance_HLL	3.5450	1.64500	7.638	0.0012310	1.2655
SmallDependenceHighGrayLevelEmphasis_HHH	0.2133	0.09202	0.4942	0.0003138	−1.5453
PFS					
SmallDependenceHighGrayLevelEmphasis_HHH	0.3715	0.2065	0.6686	0.0009571	−0.9901
TotalEnergy_HHL	2.4610	1.3910	4.3550	0.0019840	0.9006
JointEntropy_HLL	0.4969	0.2943	0.8388	0.0088420	−0.6994

OS―overall survival; LRFS―local relapse-free survival; PFS―progression-free survival.

## Data Availability

The datasets used and/or analyzed during the current study are available from the corresponding author on reasonable request.

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
