# Peer review of "Establishment of a Prediction Model for Overall Survival after Stereotactic Body Radiation Therapy for Primary Non-Small Cell Lung Cancer Using Radiomics Analysis"

_cancers, 2022, doi:10.3390/cancers14163859_

Round 1

Reviewer 1 Report

Manuscript Number:

Title: Establishment of a Predicting Model for Overall Survival after Stereotactic Body Radiation Therapy for Primary Non-Small Cell Lung Cancer Using Radiomics Analysis

Article Type: Article

Section/Category: Submission for a regular issue

General comments

The author presents a very interesting study of 358 patients divided in two group, one group is used for training, and the other for validation. Great hopes are made on the radiomics, and it is probable that we are at the beginning of their use in a clinical way

It is a single-institution retrospective study.

Material and Methods

P03L093 : The author explain that the CTV is made with a constant margin in each direction but when you breathe you do not breathe in all directions. For example, it can be considered the tumour is moving more in a head-foot axis than left-right?

Or a tracking system is used?

 P03L104 : please insert average number median number maximum and minimum

P03L105 : What is the SUV used to diagnose a recurrence?

P03L116 : Can you explain the method used to defined the ROI

Results

P5L106 : Surprisingly, the total dose is not mentioned, did it not influence the response?

Results

P088L241 : Radiomics are extracted from a manual degradation, does this delineation have an influence on the results?

Conclusion

It is an interesting paper with a large number of patients. The information given can help

We suggest accepting this paper with minor revision

Author Response

Dear reviewer 1,

Please find attached the revised manuscript, which we would like you to consider for publication. First of all, we would like to thank you for your thoughtful comments. The following revisions have been made according to these comments:

Extensive editing of English language and style required

Answer: We asked Editage careful English proofreading and corrected our incorrect expressions.

P03L093 : The author explain that the CTV is made with a constant margin in each direction but when you breathe you do not breathe in all directions. For example, it can be considered the tumour is moving more in a head-foot axis than left-right?

Or a tracking system is used?

Answer: We are sorry for our inappropriate expression. We corrected the explanation of target definition in the 2nd paragraph in “2.2. Radiotherapy” section of the Materials and Methods. Actually, we used 4D-CT to cover the respiratory motion of the tumor and did not used tracking system.

 P03L104 : please insert average number median number maximum and minimum

Answer: We inserted range of age, maximum tumor diameter, and total radiation dose of the training cohort in the 1st paragraph of in “3.1. Patients Characteristics” section of the Results.

P03L105 : What is the SUV used to diagnose a recurrence?

Answer: We recognized local recurrence when SUV-max of the tumor was higher than 2.5. We wrote this in the 2nd paragraph in “2.3. Clinical Endpoints” section of the Materials and Methods.

P03L116 : Can you explain the method used to defined the ROI

Answer: We wrote the rule of the GTV delineation at the end of “2.4. CT Image and Tumor Contouring” section of the Materials and Methods. We inserted the method of image processing before textural feature extraction in “2.5. Radiomic Feature Extraction” section of the Materials and Methods.

P5L106 : Surprisingly, the total dose is not mentioned, did it not influence the response?

Answer: We added the results of univariate analysis of total radiation dose in Table 2. It did not affect survival outcomes in this study.

P088L241 : Radiomics are extracted from a manual degradation, does this delineation have an influence on the results?

Answer: The fact that all GTVs were manually contoured may have reduced reproducibility of this study, while homogeneity of this study was kept since all contouring tasks were conducted by a single expert radiation oncologist. We wrote this in the last of the 4th paragraph of the Discussion as one of the limitations.

Sincerely,

Subaru Sawayanagi, Hideomi Yamashita

Reviewer 2 Report

1.The author's assessment of the patient's tumor background is not comprehensive enough, and there is a lack of follow-up treatment options, which will greatly affect the patient's prognosis.

2.The authors' conclusions have little clinical significance and need to continue to be discussed

3.Some references need to be updated.

4.Lack median follow-up time

5.The graph is not refined enough

Author Response

Dear reviewer 2,

Please find attached the revised manuscript, which we would like you to consider for publication. First of all, we would like to thank you for your thoughtful comments. The following revisions have been made according to these comments:

1.The author's assessment of the patient's tumor background is not comprehensive enough, and there is a lack of follow-up treatment options, which will greatly affect the patient's prognosis.

Answer: We added the data of histology, tumor site, C/T ratio, and salvage treatment in the 1st paragraph in “3.1. Patients Characteristics” section and the 1st paragraph in “3.2. Univariate Analysis” section of the results, Table 1, and Table 2. We wrote that patients with or without salvage treatment were included in the 1st paragraph in “2.1. Patients” section of the Materials and Methods.

2.The authors' conclusions have little clinical significance and need to continue to be discussed

Answer: We added the importance of prediction of recurrence risk before treatment in the 2nd paragraph of the discussion and the 1st paragraph of the conclusions.

3.Some references need to be updated.

Answer: We added current references about outcomes (reference number 13, 14) and prognostic factors (reference number 19, 20) of SBRT for early stage NSCLC. We also added the sentence about prognostic factors in the 2nd paragraph of the Introduction.

4.Lack median follow-up time

Answer: We wrote the median follow-up time for patients who survived at the last follow-up at the last of the 1st paragraph in “3.1. Patients Characteristics” section of the Results.

5.The graph is not refined enough

Answer: We changed the layout of Figure 2 to be the same as Figure 1.

Sincerely,

Subaru Sawayanagi, Hideomi Yamashita

Round 2

Reviewer 2 Report

Congratulate to the author, this paper can be accepted for publication.